# Family rejection of non-hetero sexuality– Sexual orientation and behavior anonymity among sexual minority men in slum communities-BSGH 001

**Osman Wumpini Shamrock**[1,2,3]*, **Gamji Rabiu Abu-Ba'are**[1,2,3,4,5], **Edem Yaw Zigah**[2,3], **Amos Apreku**[3,6], **George Rudolph Kofi Agbemedu**[2,3], **Donte T. Boyd**[7], **Gideon Adjaka**[8], **LaRon E. Nelson**[5,9]

1 School of Nursing, University of Rochester Medical Center, University of Rochester, Rochester, New York, United States of America, 2 Behavioral, Sexual, and Global Health Lab, School of Nursing, University of Rochester Medical Center, University of Rochester, Rochester, New York, United States of America, 3 Behavioral, Sexual and Global Health Lab, Jama'a Action, West Legon, Accra, Ghana, 4 Department of Public Health Sciences, University of Rochester Medical Center, University of Rochester, Rochester, New York, United States of America, 5 Center for Interdisciplinary Research on AIDS, Yale University School of Public Health, New Haven, Connecticut, United States of America, 6 Department of Population, Family and Reproductive Health, School of Public Health, University of Ghana, Accra, Ghana, 7 College of Social Work, Ohio State University, Columbus, Ohio, United States of America, 8 Hope Alliance Foundation, Accra, Ghana, 9 School of Nursing, Yale, Connecticut, United States of America

* osmanwumpini_shamrock@urmc.rochester.edu

**Data Availability Statement:** All data are in the manuscript.

## Abstract

It can be challenging for sexual minority men (SMM) to decide whether or not to disclose their sexual orientation or behavior. The implications of this decision are significant, especially when considering how their family might react. We interviewed individuals living in slum communities (n = 12) in Accra and Kumasi, Ghana. Our study found that two factors primarily influenced the decision of SMM to disclose their sexual orientation. Firstly, SMM feared facing harm from their families and, secondly, the close ties of SMM families to religious institutions in their communities, which taught against LGBTQ+ activities in the country. These findings contribute to understanding why SMM in Ghanaian slum communities choose to keep their sexual orientation anonymous. While no single intervention is enough to address the challenges associated with coming out, participants in the study agreed that a social support intervention that provides opportunities to educate and inform their families and community on LGBTQ+ activities could help them assimilate comfortably in their communities.

## Introduction

Many sexual minority men (SMM) worldwide face discrimination and stigma, leading them to conceal their sexual orientation and behavior [1–4]. Sexual anonymity refers to the act of concealing one's non-heterosexual orientation. or behavior out of shame, fear, or pressure to

**Funding:** The Yale University Funds for Lesbian and Gay Studies (FLAGS) grant funded this study, with additional support from the Center for Interdisciplinary Research on AIDs (CIRA) at Yale University, and the Inaugural Harriet J. Kitzman Endowed funds at the School of Nursing, University of Rochester. Funding was secured by the GRA. The funders had no role in study design, data collection and analysis, decision to publish, or preparation of the manuscript.

**Competing interests:** The authors have declared that no competing interests exist.

conform to societal gender norms from friends, family, and communities [5,6]. Acts of anonymity by SMM can cause them to experience negative emotional reactions, anticipate negative changes in their relationships, desire to maintain their current perception, and fear rejection due to cultural and religious directives [7]. SMM who do not disclose their sexual orientation/ behavior may experience poor mental health, internalized homophobia, lower self-esteem, and poorer health-seeking behavior [8–11]. Due to these asserted views from others, we infer that nondisclosure continues the suffering of identity uncertainty [4,12–16]. The effect of nondisclosure of sexual minorities' status has been linked to the family and its intertwined nature with their religious systems and practices [17–19]. Orthodoxy likely contributes to the nondisclosure of non-heterosexual sexual orientation/behavior among SMM and the family responses to the coming out process [19–21].

While sexual minority orientations and behaviors are more accepted in the Western world, many developing countries, like Ghana and sub-Saharan Africa, remain unsafe for those belonging to a sexual minority group [8,22,23]. SMM in Ghana face widespread discrimination based on their sexual orientation/behavior [23–28]. The country's laws perpetuate negative attitudes toward SMM [29–31]. Although there is no specific law outlawing non-heterosexual orientation or behaviors, many people view these activities as illegal. Such perception is driven by the criminal code under section 104, which indicates that consensual and non-consensual unnatural carnal knowledge are felonies or misdemeanors [31]. Due to the legal implications in Ghana, many SMM refrain from disclosing their non-heterosexual orientation or behavior out of fear of being criminalized [24,32–34].

Recent discussions in sub-Saharan Africa and Ghana have focused on healthcare workers' role in encouraging nondisclosure of sexuality or non-hetero behaviors among SMM [31,35–38]. Studies have investigated how sexual minorities conceal their orientation/behaviors in healthcare settings to avoid stigma and have shown that high stigma at the healthcare facility level towards SMM leads many SMM not to disclose their sexual orientation/behavior [19,31,35,38]. For instance, a study show that about 29% of health workers in Ghana would not provide services to SMM, if given the opportunity [39]. Additionally, studies in Ghana among healthcare facility staff indicate 56.4% and 60.6% of healthcare workers held fears of acquisition of HIV through SMM, and used unnecessary preventive measures for infection when they cared for SMM [34]. Thus resulting to recent stigma reduction interventions mainly targeting healthcare facilities and providers in Ghana [34,40].

Currently, in Ghana, an anti-LGBTQ+ bill named "Promotion of Proper Human Sexual Rights and Ghanaian Family Values, 2021" has been proposed as a legal instrument to control the activities of individuals in the community [41]. Presently, this bill is undergoing review in the parliamentary processes, and if passed will create an unsafe environment for persons who identify with the LGBTQ+ community [42]. The Ghanaian anti-LGBTQ+ bill has been described as a legal way of cracking down on homosexuals in Ghana and creating an atmosphere where LGBTQ+ people face threats of harm from civilians and law enforcement [43]. As of the time this paper was drafted, the anti-LGBTQ+ bill in Ghana is being pushed to be passed as a law by a section of members of parliaments and religious bodies [8,44,45].

It is imperative to draw attention to the lacking aspects of the life of SMM in geographical locations as well as driving factors that facilitate nondisclosure of their sexual orientation/behavior. It is important to understand the impact of family on sexual orientation/behavior among SMM in Ghanaian slum communities as stigma leads to anonymity, psychological stress, poor health-seeking behavior, and marginalization of this population [8,13,14]. Studying the influence of family on the anonymity of sexual orientation and behavior among SMM in Ghanaian slum communities can help develop strategies for "coming out" and reducing stigma.

### The importance of this study in Ghanaian slum communities

Slum communities remain characterized by poverty, unemployment, poor and inadequate housing, overcrowding, and insufficient essential services such as electricity, water, waste collection, health, and educational facilities [23,46,47]. In Ghana, an estimated 37.4% of the population resides in slums. yet, development projects have largely overlooked these communities, resulting in substandard living standards [48,49]. The number of people living in urban slum communities in Ghana rose from 5.5 million in 2017 to 8.8 million in 2020 [50]. While estimates of SMM living in Ghanaians slums remain unknown, a recent study shows that SMM in Ghana's urban slums faces difficulties accessing healthcare due to challenges in disclosing health information and sexual orientation, navigating healthcare, and fear of stigma from familiar faces or healthcare providers [8].

Like elsewhere, individuals living in slum communities in Ghana, due to their low educational status, may tend to associate sexual behavior with procreation rather than personal preference [46]. The high rate of poverty, unemployment, and inadequate housing in slums results in communal living, as people in these communities may prefer to live with family and friends to offset the high cost of living [51]. The family, which serves as a primary agent of socialization in Ghana, may play a significant role in influencing sexual behavior and outcomes [52–56]. However, previous studies in the country do not capture the influence of family on sexual behavior in the life of SMM, especially for those in slum communities. Given these statistics and features of slum communities in the country and the lack of information on the nondisclosure of sexual orientation/behavior, we aim to explore the impact of family on SMM disclosure of their sexual orientation/behavior in the selected slum communities. The study will aim to unearth the experiences of SMM, which maybe be vital in public policy intervention. The study will seek to provide support to theories such as the Minority Stress Theory in understanding SMM's external and internal stressors and stress amplifying conditions among this population in the Ghanaian context.

## Methodology

### Research design

The study employed a qualitative phenomenological design approach to understand and describe the lived experiences of SMM in Ghanaian slum communities. The approach allowed us to gather firsthand narratives from participants about their everyday lives as SMM within the larger family systems in slum communities and how they interpret their experiences concerning their sexual orientation/behavior. Using a phenomenological approach allowed us to employ in-depth, open-ended interviews with SMM to describe their lived experiences on disclosure, rejection, or acceptance. The phenomenological nature of this study also allowed for the use of direct quotations from SMM to provide a vivid description of their experiences and to maintain the richness and depth of the participant's narratives. Considering the subjective meaning and interpretation from SMM, researchers were able to analyze data by focusing on understanding the participant's perspectives on family rejection, sexual and behavioral anonymity, and how these factors influence their everyday lives in the slum community.

### Participants

A total of 12 participants were sampled from slum communities in Accra and Kumasi. All participants were 18 years and above at the time of the interview. Participants were born male and identified as cis-gender men. All participants confirmed having sex with another male within the past six months. Four participants indicated being Muslim, six as Christians, and two as

Muslim and Christian. Five participants had tertiary education, and six completed Senior High School education. One participant did not complete Junior High School education.

## Sampling and recruitment procedure

In collaboration with our community partners in Accra and Kumasi, we used time-location sampling (T.L.s), a type of convenience sampling technique to reach and recruit SMM in slum communities [57]. This sampling method was employed to assess SMM in the targeted slum communities conveniently. Research assistants working with our community partner organizations in Accra (Priorities on Rights and Sexual Health–PORSH) and Kumasi (Youth Alliance on Health and Human Rights–YAHR) screened and invited SMM to participate in interviews sessions during one of the organizations' activities when SMM visited the location. We have a long history of working with these two organizations in recruiting and implementing studies among SMM in Ghana [8,24]. We initially targeted to recruit 19 participants, but we reached saturation in responses around the 8th interview; we continued to gather an additional 4 to ensure full information saturation, making our total transcripts 12.

## Inclusion criteria

Individuals were invited to participate in the study if they were at least 18 years of age and lived in a slum community in Ghana's major urban areas of Kumasi, Ashanti region, and Accra, Greater Accra region. The individual must also self-identify as a cis-gender man and belong to an SMM category (gay, bisexual, pansexual, or have sex with other cis-gender men for reasons than sexual orientation). The person must also be sexually active and reported having sexual intercourse with another cis-gender man within the last six months before speaking with research assistants about this study.

## Data collection procedure

**Procedure.**  Typical of the phenomenological design, we used in-depth face-to-face interviews to solicit participant data [58].Upon completion of screening, research assistants handed over consent forms to participants to read. The research assistants read the consent forms aloud and provided further explanations to ensure clarity. They also responded to questions from participants and collected their signatures as consent for participating and allowing audio recording before proceeding with the interviews. All interviews occurred in secured locations of the community partners. Four interviews were recorded in Twi, a Ghanaian language, as some participants could not effectively express themselves in English; the remainder were in English.

**Nature of questions.**  The research assistants received training on qualitative interviews using the checklist generated for the study. Consistent with our design, the checklist focused on allowing free and open conversation rather than a typical question-and-respond interview process. Participants were asked to provide personal narratives about their background and family, sex and gender expectations, experiences of stigma within the family, openness about sexuality or sexual behavior, acceptance in the family, and how they cope with stigma or live as SMM.

**Analytical strategy.**  Trained research assistants transcribed the audio interview recordings verbatim and deidentified the transcripts by removing specific descriptions that could allow others to identify the participant. We then subjected the transcripts to a multiple-reviewer summative content analysis process [24]. Our team has successfully used this analytical process to understand key factors in participant accounts [24]. We assigned each transcript to at least two reviewers. Each reviewer examined the interview checklist and then reviewed

**Table 1. Major themes and categories developed.**

| Major Theme | Categories |
|---|---|
| Theme 1: The fear of facing harm from family | Fear of physical assault from family's perception of SMM activity. |
| | Fear of media |
| | Fear of deviation from community culture and gender norms, and expectations |
| | Criminalization of SMM activities by LGBTQ+ legislation |
| Theme 2: Anonymity influenced by religion. | Anonymity of SMM due to religious affiliation (Christian/Muslim). |
| | Religious teachings against SMM contributed to them being anonymous. |

*Notes.* Table 1 demonstrates the themes and categories identified during this study's analysis and coding process.

the transcripts to identify the most salient factors raised by the participants, which they independently reported using between 100 to 200 words. The lead author reviewed all summaries and organized the salient points from each write-up into a data spreadsheet, which guided the identification of clusters in the qualitative data and showed the factors that frequently appeared in the transcripts and summaries (Table 1). We identified several clusters, called categories, and put them under broader themes that explained participant experiences within the family and how they navigate sexual stigma within the families. Each category reported appeared as a salient factor among both peer reviewers.

## Ethical considerations

The study received ethics approvals from the Institutional Review Board Committee at Yale University in Connecticut, USA (approval number IRES IRB #RNI00002010) and the Ghana Health Service Ethics Committee in Ghana (approval number GHS-ERC 001/10/21). The interviewers involved in the study made sure that all participants had a clear understanding of the informed consent and obtained written approval prior to collecting any data.

## Results

### Description of themes and categories

Two themes emerged from the data analysis that captured family-related factors that influenced SMM attitudes and forced them to stay anonymous in their communities. The first theme was the fear of facing harm from the family. Categories within this theme were 1. Fear of physical assault from family, 2. Fear of media, 3. Fear of deviation from community culture and gender norms and expectations, 4. Criminalization of SMM activities by LGBTQ+ legislation. A second theme in the analysis was anonymity influenced by religion. Categories in the second category were: 1. Anonymity of SMM due to religious affiliation (Christian/Muslim), 2. Religious teachings against SMM contributed to them being anonymous.

### The fear of facing harm from family

Fear emerged as a cross-cutting theme for SMM in the study. Within this theme was the fear of harm and physical assault from the family of SMM, the fear of media misinformation creating a hostile environment for SMM, the fear of deviation from community culture and gender norms and expectations, and the fear emanating from the criminalization of LGBTQ+ activities by the law.

## Fear of physical assault from family's perception of SMM activity

Some participants indicated they feared physical harm from their family and relatives if their sexual orientation/behavior were ever revealed. In-depth interviews with SMM revealed other external factors (e.g., media and individuals outside their families) who harbored ill feelings towards the idea of LGBTQ+, contributing to the anti-LGBTQ+ stance of their families. Threats to SMM by their families in the study included poisoning and physical beating.

> I am cool with my family. They do not know I am MSM. Even if they know, no one has caught me in the act or seen me mingle with MSM people because of our Muslim community; they see MSM people as devils. For instance, my older sister recently passed a comment after listening to LGBTQ issues on the radio. She said that if any member of her family is caught as an LGBTQ member, she will personally poison that family member. I didn't say anything because I wasn't ready for the confrontation (SMM participant).

The negative perception surrounding SMM in slum communities contributed to a situation where their families threatened to end their lives or subject them to physical pain to deter the practice of any form of LGBTQ+ activity. SMM stated adopting innovative protective mechanisms; reducing the number of acquaintances or friends was a sure way of staying anonymous to prevent the family from knowing about their sexual orientation/behavior and being subjected to physical harm. Religion was also cited as a social factor that prepared the right community atmosphere to fuel hate for SMM in this study. Given the above cited quote, the role of religion had a cross-cutting influence on other aspects of the community, mainly through radio, which served as a medium to spread hate for SMM in slum communities.

## Fear of media

SMM mentioned media as a medium through which misinformation is used to spread hate among their family members and fear among SMM. Speaking specifically on the role of media, one SMM mentioned, "I believe the media is feeding the society with false information about the community." Another SMM in the study indicated despite the ill-spoken tag for SMM in society, they stood firm with their identities. One of the SMM in the study stressed:

> With this LGBTQ thing going on in Ghana, the last time I watched these human rights on GTV (a local Ghanaian T.V. station), they were talking about LGBTQ stuff. We didn't ask God to put this thing in us. Because some of us were born with it naturally, some of us were taught by friends and other stuff. But in the end, we are the same people. God created us and he created us in his image. So, the only thing they should stop doing is, they should allow us to be free and move along with what we feel like doing (SMM participant).

Stressing the role of media, another SMM mentioned:

> For instance, my older sister recently passed a comment after listening to LGBTQ issues on the radio. She said that if any member of her family is caught as an LGBTQ member, she will personally poison that family member (SMM participant)

These assertions expressed by SMM showed the role of media as a facilitator of stigma and how this connects to the family structure. The worry expressed by SMM in these interviews showed media as a tool that contributed to brainwashing the community and SMM families'

members about the harm associated with being a member of the LGBTQ+ community. Media in this study was not mentioned in relation to spreading education or acceptance of SMM.

Expressed intent to cause harm by family members in this study formed a driver that facilitated anonymous behavior by SMM in slum communities. Despite the threat of harm and physical abuse from individual family members, some SMM indicated they would attempt coming out if they could secure financial security from living under the roof of their family or if they could possess some items of wealth securely. According to one of the SMM in the interview, "I will come out when I think they understand me; that's why I am working hard. I believe that when one day I am successful, they will have no choice but to understand me" (SMM Participant). This statement indicates that being financially stable and not reliant on family meant SMM could openly express their sexual orientation/behavior to their families without feeling unsafe.

## Fear of deviation from community culture, and gender norms and expectations

The dominant Ghanaian culture and norms, which criminalizes non-cisgender and non-heterosexual orientation/behaviors, and gender expressions, were deep-seated beliefs of inhabitants within the slums. As a result, SMM individuals did not want to be perceived as feminine or associate with men who expressed feminine features, as they strived to stay anonymous. SMM in non-heterosexual relations indicated they deliberately tried diverting attention by presenting a masculine gender expression to remove all forms of doubt from their community and family members. SMM indicated feminine gender expressions were not shared culturally by most members of their communities, including their respective families. In discussing the existing community culture in the slum community and the motivations to remain anonymous as an SMM, one of the individuals stated:

> I haven't had any such experiences. I haven't opened up to anyone in my community. About whom I am. So, they don't know. I don't have feminine features and I also don't have feminine friends. And so, they don't know. Sometimes if you work with feminine guys, they will think that you are also like that. And they will think something bad. I don't mingle with my community friends. I mingle with outsiders far away from my community. Because it's my secret (SMM Participant)

Another respondent stressing the safety of SMM in a community with firm heterosexual beliefs and why they chose to stay anonymous mentioned:

> My community members don't know that I have sex with men. But except the ones I have something to do with who already know. But they don't usually call me by name. My community is a place where if they get to know you are MSM the society frowns on you. So, to be on the better side, we hide what we do. But let's say my community is not a good place for me to live, but for safety's sake, I need to pretend and be myself (SMM Participant).

Remaining anonymous within a community with firm heterosexual beliefs for SMM was also connected to negative perceptions of oneself from close acquaintances, particularly from family. According to respondents in the study:

> I find it difficult being a Muslim in my community. I do my stuff on a low key. I don't show my identity to my community members, friends, people and family. They do not know the real me. They know that I'm a normal guy. They do not know I'm MSM. I don't let them

know about me because I feel my family won't be okay with knowing who I am (SMM Participant).

Similarly, another SMM in the interview mentioned:

My community members don't know about my sexuality because it's unsafe to reveal your sexuality in our area. Because the Christians believe that it's not right and my dad also believes it's not right. When I was younger, my dad would always complain about me being girly. So, I had to force myself to behave like a man. So, in my area, we don't open up about our sexuality (SMM Participant).

While heterosexual tendencies within the slum community dictated the existing culture around SMM activities, and closely associated with religion, some SMM indicated their family's concerns superseded all other reasons for staying anonymous. According to one of the SMM, "It's not about religion for me; it's about my family members. That's why I have not come out. I will come out when I think they understand me; that's why I am working hard." These assertions concerning community culture contributed to why SMM decided to stay anonymous, with the family element emerging as one of the primary reasons.

## Criminalization of SMM activities by LGBTQ+ legislation

During interview sessions, SMM mentioned that their decisions to stay anonymous were also informed by the legal instruments in the country that portrayed their activities as not conforming to normal social behavior. According to SMM in the study, the criminalization of their activities served as a deterrent and a facilitator of hate within the slum communities. Many of the participants mentioned members of their communities and families used existing and ongoing legislation against LGBTQ+ as a reason to continue propagating hate against individuals who practiced or were suspected of being SMM. Speaking on ongoing efforts to push for additional bills against LGBTQ+ in the country, one of the participants mentioned, "for me, the bill is irrelevant for people to try and put up. Because what someone does with another's consent is no one's business." According to this SMM, the push for additional bills in the Ghanaian parliament was unnecessary and was not in the interest of anyone. Another SMM speaking on the consequences of such bills and existing laws mentions the implications of these legislations:

I do it secretly, so my sexuality is unknown to my community. Sometimes my friend will talk about how they will treat anyone caught in that act. Because I am into guys, I always listen to them closely and ensure they don't know about my sexuality. I always sneak from them, attend to my clients, and sneak back to them without knowing my movement. If they get to know who I am, they can harm me, so I am always careful (SMM Participant).

According to SMM in the study, the continued existence of existing legislation and those in the pipeline facilitated their decision not to disclose their sexual orientation/behavior. Other SMM in the study indicated the unfair nature of the bill as one that was misdirected and of no essence to the country's current needs. According to one of the SMM:

They are not being fair to us because there is nothing that they can do to change who we are; besides, there are a lot of things going on in the country that affects the development of the country, but they are overlooking all these and focusing on us which is not fair (SMM Participant).

SMM indicated the essence of legislative laws and bill in the country only facilitated their unfair treatment. Participants also expressed worry about how the existing laws and bills created an atmosphere of fear which prevented them from disclosing their identities to family members and friends. Stressing nondisclosure, one SMM mentioned:

> When I heard about the bill, I was very scared. I didn't know what to say because you can't even tell your friends about it, your straight friends and family. When you try to defend LGBTQ, the community will begin to suspect you. So, you have to keep quiet. But within me, I know the bill isn't going to help us at all (SMM Participant)

The continuous push for legislation against LGBTQ+ through the current anti-gay bill "Promotion of Proper Human Sexual Rights and Ghanaian Family Values, 2021" created an atmosphere of homophobia against the community and was cited by SMM as a reason they chose not to disclose their sexual orientation/behavior to friends or families. SMM also mentioned public advocacy against passing the bill were acts that put them in harm's way. The existence of the bill created a criminal public perception of LGBTQ+, pushing SMM to keep their sexual orientation/behaviors anonymous.

## Anonymity influenced by religion

SMM in the study indicated that being anonymous about their sexual orientation/behavior was closely influenced by their religious affiliations and teachings. During interview sessions, SMM in the study mentioned belonging to either Christian or Islamic faiths, or both. Participants mentioned family religiosity and their religious affiliations prevented them from expressing their sexual orientation/behavior publicly. Participants also mentioned religious teachings in their respective religion of choosing, preached against their sexual orientation/behavior contributing to them staying anonymous.

## Anonymity of SMM due to religious affiliation (Christian/Muslim)

Respondents indicated their family's affiliation and religious stance on LGBTQ+ activities forced them to stay anonymous to prevent relatives from discovering their sexual orientation/behavior. A reason for remaining anonymous was that other community members belonging to the same religious groups as their family could easily reveal their identities. Respondents also mentioned they stayed anonymous to prevent disappointing their family members affiliated with Christianity or Islam religions. One of the respondents indicated:

> I don't show my identity to my community members, my friends, and my family. They do not know the real me. They know that I'm a normal guy. They do not know I'm MSM. I don't let them know about me because I feel my family won't be okay with knowing who I am. Mostly my mum will be angry with me and feel very disappointed. I'm not open to my family members. But to the community, I really do not care (SMM Participant).

SMM also indicated leadership roles occupied by their family member in religious organizations reinforced their action targeted at staying anonymous. One of the SMM interviewed mentioned that his father's role within the Muslim organization in the community forced him to remain anonymous and will only attempt to reveal non-heterosexual orientation/behaviors to fellow SMM who lived outside his community or seek health-related concerns in a location far from his community. One SMM mentioned, "My father is like an Imam, so I have to comport myself. I don't want to disgrace my family. I should not let my actions tarnish the image

of another person". The power of religious affiliation in this study indicated SMM preferred to stay anonymous to protect the religious image of their family or the positions they held within religious organizations in the community. Religious leaders in slum communities according to participants, did not condone persons acting in ways that identified with LGBTQ+ or people who supported the practice.

### Religious teachings against SMM contributed to them being anonymous

Religion was also cited as an instrument to sensitize the community on how not to engage in LGBTQ+ activities and the spiritual harm that came to people who engaged in such practices. Religious leadership was mentioned as agents through which these narratives were pushed forward to the community members in slum societies. According to an SMM in this study:

> When I go to church, they will be preaching against me. It's like the pastor is speaking against me. So, I don't like going, but when I feel like it, I will go to church, but if not, I will sleep. The pastors want to condemn gays and see it as a big sin and they themselves are sinning on a low key. Sometimes when I'm there and they preach, I become sad, but I know I didn't choose this for myself. It was when I was in SHS that sometimes I become shy when I go somewhere. But as of now, I don't care (SMM Participant).

Given this statement, SMM whose family relatives were members of the clergy in the community knew about the importance of these family relatives' role and took precautionary steps to prevent shame and embarrassment to these family relatives. Another participant mentioned, "I don't attend church because of my sexual orientation." SMM cited the decision to stay away from places of worship to keep anonymous from families and the community. Together anonymity for SMM in slum communities meant avoiding harm from family and, at the same time, a step towards preventing shame and embarrassment due to their sexual orientation/behavior.

### Discussion

This study utilizes a qualitative phenomenological design to understand how family rejection of SMM sexual orientation/behavior influenced their decision to stay anonymous in Ghanaian slum communities. Prior research on the role of the family in the lives of SMM has provided little insight into why they choose to remain anonymous. Anonymity for SMM has far-reaching consequences that are not only physical but psychological for SMM. This study provides sufficient background to fill the gap in knowledge on the role of family and why SMM decide to remain anonymous in Ghanaian slum communities. This study revealed the role of families and how their knowledge of SMM sexual orientation/behavior contributed to them staying anonymous. The study's results show that two factors drove SMM to remain anonymous. The study's first emerging concern for SMM was the fear of facing harm from family relatives from disclosing their sexual orientation/behavior. The second was SMM's anonymity due to religious organizations' role within the community and how these institutions facilitated the nondisclosure of their sexual orientation/behavior.

SMM face physical threats from their family members if their sexual orientation/behavior are revealed or if they are engaged in LGBTQ+ activities. According to the SMM, these threats were motivated by external factors such as media, the laws on LGBTQ+ and the negative influence from close relatives who harbored ill feelings towards their activities. Participants described their sexual orientation/behavior as detrimental to their safety. SMM commonly cited experiences of physical threats from the family in the study as one that kept them from

coming out. However, none of the interviewees mentioned ever experiencing physical attacks from their families. StudiesClick or tap here to enter text. that focus on SMM agree that family relatives were among the largest perpetrators of threats and actual physical harm among individuals whose sexual orientation/behavior differed from socially acceptable gender roles and standards that aligned cisgender and heterosexual orientation/behaviors [59]. Other studies have corroborated claims of sexual and gender violence from families and cited countries within Africa's Sub-Saharan region to be high on this occurrence [60,61]. While we identified evidence that SMM avoided disclosing their sexual orientation/behavior to family due to the perceived risk of facing harm from their relatives, we also recognize that SMM may avoid disclosure in their communities to prevent attacks on their family members [62,63].

Similarly, the decision to remain anonymous for fear of family rejection has been cited [64] to show people in same-sex relationships left home early due to differences in ideological thoughts between themselves and their parents. Findings in this study highlight conflicts in thoughts and unfriendly community atmospheres as drivers that facilitated SMM moving out of the family setting. Also, participants in this study indicated in several interviews the decision to stay anonymous for fear of disapproval of their sexual orientation/behavior from parents. Studies show parents who knew their children's sexual orientation/behavior were more likely to reject these identities which resulted in the dissolution of their relationships [65]. Prior research has also found that family rejection of identity is detrimental to SMM's mental and physical health [66–69].

This study reinforces these assertions, highlighting the role of the family as a threat and enabler that prevents SMM from coming out in their respective societies. Our results acknowledge the role of dominant Ghanaian culture and norms, which criminalizes homosexuality and encourages heterosexual relationships supporting assertions from previous studies [70]. The need to divert attention and stay anonymous by acting masculine and avoiding feminine men was a developed mechanism by SMM in our study to show conformity to community gender and sexual expectations (cisgender and heterosexual orientation/behaviors), which were not culturally shared by the majority of people, including their families. The study also reemphasized the effects of SMM keeping anonymous their sexual orientation/behavior and the implications on their social environment as captured by others [71–74]. The importance of addressing these challenges by SMM and their family's role in how their lives are experienced in the community has important implications for policy consideration. A previous report published by the Solace Brother Foundation, The Initiative for Equal Rights, The Center for International Human Rights, and The Heartland Alliance for Sexuality and Human Rights specifically mentions the family as a factor that individuals take into consideration when showing their and sexual orientation/behavior because of the potential negative implications for their relatives [75–77].

Despite citing the family as a facilitator of fear that prevented SMM from disclosing their sexual orientation/behavior, other studies have emphasized different views and mention the family structure as a tool used by SMM to cope with challenges associated with disclosing their sexual orientation/behavior. Several studies have shown the involvement of the family in the lives of SMM facilitated safer sexual practices and supported the mental health and wellbeing of these individuals [78–81]. Studies have show the family's involvement in an SMM's life has been shown to be supportive when dealing with sexuality-related issues [79]. While this study shows family rejection as a critical factor in coming out for SMM, it also highlights the role of place (Ghanaian slum communities) as a factor to consider that may influence support or refusal of sexual behavior.

Given the findings of this study, it is important to understand and propose an intervention to curb the adverse effects of parental rejection of SMM sexual orientation/behavior. For SMM

in Ghanaian slum communities, parental knowledge was key in confronting sexual and gender-related identity struggles. Specific social support interventions can be identified by identifying family rejection and its associated challenges for SMM to create an opportunity for sexual minorities to assimilate comfortably in their communities.

This study also establishes that factors within the socioeconomic context such as self-reliance and being financially autonomous from the family contributed to why SMM kept anonymous their sexual orientation/behavior. A typical trajectory for many SMM was relying on family for some of their social or economic needs. Recognizing the role of socioeconomic context in the lives of SMM, some studies [82] support proposals to create an intervention to provide socioeconomic relief for SMM and consider contextual social diversity. Discussions of religion came up as a driver of stigma as many SMM mentioned religiosity as the reason they were stigmatized by their family and community members. Over the years, religion has been captured in scholarship, specifically focusing on how the body contributes to psychological and emotional stress for sexual minorities [83–86]. Studies have highlighted the role of religion in spreading homophobic messages that have been found to significantly affect the lives of sexual minorities in their communities and the role of the body in contributing to internalized homonegativity and hate [86–88]. Within Sub-Saharan Africa, religion has been identified as a factor that could adversely affect how communities treat SMM [19]. While this study highlights the role of religion in the lives of SMM, it narrows down how this phenomenon plays out in Ghanaian slum communities and how the family role in these institutions contributes to increased discreet sexual behavior. Interventions to address the role of religion as a contributing factor to the adverse experience of sexual minorities in this study fall in line with those studies which focus on public sensitization [19,86].

Several studies stress the importance of focusing on slum communities in Sub-Saharan African countries[23,89–91]. Given the numerous challenges faced by inhabitants of slum communities, others have stressed the importance of focusing on sexual minorities living in these communities [92,93]. In line with this call, this study's focus on slum communities revealed the growing concerns surrounding the role of the family in the lives of SMM and the challenges attached to revealing their gender and sexual identities in Ghanaian slum communities. Interventions in this study align with calls [93] to push for community education and other forms of advocacy around human rights for sexual minorities.

Studying stigmatized groups such as SMM and the unique stressors they face can be understood through the use of the Minority Stress Theory [94,95]. This theory has been employed to study Ghana to study similar populations [96]. This study offers some insights into factors that influence the attitudes of SMM and the decision to stay anonymous in selected slum communities in Ghana and align with the concepts of the Minority Stress theory. The theme reveals themes such as the fear of harm from family, media misinformation, deviation from cultural norms, and the criminalization of LGBTQ+ activities contributed to SMM's anonymity. Together, these factors highlight the external stressors faced by SMM, and internal stressors related to the internalization of stigma and the conflict between religious teachings and their sexual orientation/behavior. The study supports the concept of the theory by laying out the unique stressors faced by SMM and their impact on anonymity and disclosure decisions.

## Implications

The study forms the basis for considering intervention for program development around the topic of disclosure and family involvement in the lives of sexual minorities. The finding in this study will provide awareness to understand the difficulties SMM face when disclosing their identities in Ghanaian slum communities. At the same time, it is important to expand the

scope of this study to provide help mechanisms to support SMM living in other parts of Ghana and Sub-Saharan Africa. Findings in the study could also serve as a basis for an expanded form of intervention that identifies resources and services for educating families and communities of sexual minorities using firsthand accounts of challenges associated with disclosing their sexual orientation/behavior.

The study also sets the stage for stakeholders to engage existing policies around stigma reduction and develop strategies that consider the family unit when developing or implementing stigma reduction strategies around sexual minorities. Given the findings in this study, stakeholders will be informed of the role of the family in the lives of sexual minorities, and whose influence could impede the achievement of national targets such as the U.S. President's Emergency Plan for AIDS Relief (PEPFAR), a strategy to prevent HIV and accelerate efforts towards reducing infection, and the United Nation's 90-90-90 initiative, a campaign to accomplish 90% of people knowing their HIV status, 90% of people will receive sustained antiretroviral therapy and 90% will be in viral suppression from antiretroviral therapy by the year 2030 [97]. We also suggest strategies to consider and implement resources for families who are in support of their relatives' sexual orientation/behavior.

Finally, sexual minorities, particularly SMM in Ghanaian slum communities will find this study useful when disclosing their sexual orientation/behavior. The study provides some information on the challenges associated with revealing their identities and suggestions on what might prove helpful to SMM when deciding to disclose their identities.

## Limitations and future research

While the study depicts the living situation of SMM around family rejection of sexual orientation/behavior using a qualitative research design to sample 12 SMM. The researchers in this study would have benefited more from a much larger sample of SMM. For future studies, researchers in this study will be curious to see how these experiences reflect in a much larger sample of SMM and other geographical locations than those in slum communities. Although the age limit to qualify participants in this study was placed at 18 and older, it did not consider SMM younger than 18 years. Researchers will be curious to see future studies engage a much younger population of SMM in Ghanaian slum communities around anonymity and family influence. It will be in the interest of researchers in this study to know if age plays a role in SMM disclosing their sexual orientation/behavior.

## Conclusion

The decision to remain anonymous among SMM varies across individual basis. However, common among these decisions is the role families play when these decisions are considered. For SMM, the fear of facing harm from the family, or how religious institutions portray their activities formed a large part of why they decide not to disclose their sexual orientation/behavior. This study acknowledges that while no single intervention is enough to curb the harmful effects of SMM disclosing their sexual orientation/behavior it is imperative to identify social support interventions to educate their families and community members and create an opportunity for sexual minorities to assimilate comfortably into their communities through disclosing their sexual orientation/behavior.

## Author Contributions

**Conceptualization:** Osman Wumpini Shamrock, Gamji Rabiu Abu-Ba'are, Edem Yaw Zigah, Amos Apreku, George Rudolph Kofi Agbemedu.

**Data curation:** Osman Wumpini Shamrock.

**Formal analysis:** Osman Wumpini Shamrock, Amos Apreku, George Rudolph Kofi Agbemedu, LaRon E. Nelson.

**Funding acquisition:** Gamji Rabiu Abu-Ba'are, LaRon E. Nelson.

**Investigation:** Osman Wumpini Shamrock, Gamji Rabiu Abu-Ba'are, Edem Yaw Zigah, George Rudolph Kofi Agbemedu, Gideon Adjaka.

**Methodology:** Osman Wumpini Shamrock, Gamji Rabiu Abu-Ba'are, Edem Yaw Zigah.

**Project administration:** Gamji Rabiu Abu-Ba'are, Edem Yaw Zigah, Gideon Adjaka.

**Supervision:** Donte T. Boyd.

**Validation:** Osman Wumpini Shamrock.

**Visualization:** Osman Wumpini Shamrock, Amos Apreku, Donte T. Boyd, LaRon E. Nelson.

**Writing – original draft:** Osman Wumpini Shamrock, Gamji Rabiu Abu-Ba'are, Edem Yaw Zigah, Amos Apreku, George Rudolph Kofi Agbemedu.

**Writing – review & editing:** Osman Wumpini Shamrock, Gamji Rabiu Abu-Ba'are, Amos Apreku, Donte T. Boyd, Gideon Adjaka, LaRon E. Nelson.

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
