## [Decision Letter · Decision Letter 0]

29 Jun 2023

PGPH-D-23-00095

Family rejection of sexuality – sexual and gender anonymity among sexual minority men in slum communities.

Dear Dr. Shamrock,

Thank you for submitting your manuscript to PLOS Global Public Health. After careful consideration, we feel that it has merit but does not fully meet PLOS Global Public Health’s publication criteria as it currently stands. Therefore, we invite you to submit a revised version of the manuscript that addresses the points raised during the review process. Thanks for your patience with the delayed review.

The reviewers have requested clarifications to the objectives as stated and regarding the terminology (e.g., gender identity vs. gender expression?) and self-identifications (e.g., MSM) used. Additional detail is needed on the methodology (phenomenology) and sampling method. Reviewer 1 has also suggested ways to improve the presentation of results (but please note that I do not support their request not to introduce new literature or references in the Discussion section). Kindly make sure to submit a clear Response to Reviewers that indicates where each revision has been made in the manuscript; this will support a quicker turnaround. 

We look forward to receiving your revised manuscript.

Kind regards,

Peter A. Newman, Ph.D

Academic Editor

Journal Requirements:

Additional Editor Comments (if provided):

Reviewers' comments:

Reviewer's Responses to Questions

**Comments to the Author**

1. Does this manuscript meet PLOS Global Public Health’s publication criteria? Is the manuscript technically sound, and do the data support the conclusions? The manuscript must describe methodologically and ethically rigorous research with conclusions that are appropriately drawn based on the data presented.

Reviewer #1: Yes

Reviewer #2: Partly

2. Has the statistical analysis been performed appropriately and rigorously?

Reviewer #1: Yes

Reviewer #2: N/A

3. Have the authors made all data underlying the findings in their manuscript fully available (please refer to the Data Availability Statement at the start of the manuscript PDF file)?

Reviewer #1: Yes

Reviewer #2: Yes

4. Is the manuscript presented in an intelligible fashion and written in standard English?

Reviewer #1: Yes

Reviewer #2: No

5. Review Comments to the Author

Reviewer #1: Thank you for inviting me to review this manuscript, which I have read with great pleasure. Anonymity/ non-disclosure and disclosure of sexual identity make an important contribution to SMM literature and advocacy efforts. However, the manuscripts would do with some review to for coherency.

Some comments for consideration;

Methodology: explain how the phenomenological approach was used, and how it has been applied to the findings.

Some themes and related analysis is not clear:

Your first theme shows why they fear. But as you move on it is about the causes of fear of disclosing /challenges of disclosing. If this is the structure, make it clear in the introduction of the findings section>

Page 13/14: Theme: Fear of deviation from culture is communicating something that is not clear. Review the section for clarity.

Page 12/13. Theme: Fear of social media- using quotes used from conventional media (radio, television, and newspapers) makes me feel you should rename the theme to Fear of the media, this way gives you the opportunity to write about any media.

Page 11/12: Theme: Fear of physical assault from family’s perception of SMM activity. Work on coherence. You mix the fears and coping strategies: I suggest, first, write about the fears, and second the coping mechanisms.

Page 15/16: Theme: Criminalization of SMM activities by LGBTQ+ legislation. This section is well-written, and clear. However, provide some brief information about the bill.

Page 16. The sentence: The country's legislative focus against LGBTQ+ through existing laws and bills did nothing to address the real challenges of the economy but facilitated the unfair treatment of SMM in the country. Do the law and bills to promise to address the real challenges of the economy?

Page 16: The sentence: The continuous push for legislation against LGBTQ+ was cited as why they chose not to disclose their gender identity and sexual orientation/behavior to their friends or families. You make a good point but needs to be unpacked.

Discussion: Try not to introduce new literature or references in the discussion section. Instead, you can introduce the new literature in the introduction section.

Reviewer #2: The manuscript addresses an understudied research topic from Ghana, with implications for similar contexts globally.

1. The authors state they “aim to explore how impoverished communities and families contribute to SMM disclosure of their gender identity, orientation and sexual behaviors”. However, the impact of poverty or living in slum communities on disclosure/nondisclosure of sexual orientation/behaviors is not explored, although the impact of family and religious affiliations on disclosure is discussed in detail. The authors can thus consider revising the aim.

2. Throughout the manuscript, the term ‘gender identity’ appears, while the inclusion criteria state that all the participants “must self-identify as a cis-gender man”. Probably the authors meant ‘gender expression’ (and not ‘gender identity’) in most contexts, even though, to add to the confusion, one participant reported, “I don’t attend church because of my gender identity”. The authors need to clarify how the term ‘gender identity’ fits within the context of conducting research among cisgender-identified MSM.

3. Some background information about the percentage of those who practice different religions can be added to the Introduction section as part of understanding the religious contexts.

4. Most of the participants refer to themselves as “MSM”, although the authors do use the term “a member of the LGBTQ community” as well. While the term “MSM” itself can be an identity, I suspect that many of the participants may have some kind of indigenous identities based on their sexual role and gender expression. In fact, one participant has been quoted to be saying “I don't have feminine features and I also don't have feminine friends. And so, they don't know”. Apparently, feminine gender expression is seen as a proxy for sexual orientation/identity. Can the authors discuss about these aspects (indigenous identities, role of gender expression in labelling someone as MSM)?

5. I could not locate any theoretical frameworks (e.g., minority stress theory) being mentioned by the authors even in the Discussion section. Although not mandatory, the authors can consider referring to appropriate theoretical frameworks to strengthen the contribution of this manuscript.

6. There are no explicit discussions on the impact of disclosure or nondisclosure on mental health of SMM.

7. The type of sampling used in this study is not explicitly specified, although it appears to be convenience sampling, with participants recruited from agencies working with MSM.

6. PLOS authors have the option to publish the peer review history of their article (what does this mean?). If published, this will include your full peer review and any attached files.

**Do you want your identity to be public for this peer review?** For information about this choice, including consent withdrawal, please see our Privacy Policy.

Reviewer #1: **Yes: **Emmy kageha igonya

Reviewer #2: No

---

## [Decision Letter · Decision Letter 1]

13 Oct 2023

PGPH-D-23-00095R1

Family rejection of non-heterosexuality – sexual orientation and behavior anonymity among sexual minority men in slum communities-BSGH 001.

Dear Dr. Shamrock,

Thank you for submitting your manuscript to PLOS Global Public Health. Your revisions largely respond to the reviewers' comments. However, please address the two outstanding comments by one reviewer, at which point the manuscript can be quickly adjudicated, and moved forward if addressed. Thus, we invite you to submit a revised version of the manuscript that addresses the 2 points raised during the review process.

Kindly attend to the outstanding comments by the reviewer, in particular in regard to your description of time-location sampling, however without any methods that appear to support that depiction. As suggested by the reviewer, please either revert to a description of convenience sampling or provide methodological details to support time-location sampling. As to the second comment, please consider if the suggestion has merit in the context of your research; and if so, you might add a brief explanation to that end.

We look forward to receiving your revised manuscript.

Kind regards,

Peter A. Newman, Ph.D

Academic Editor

Journal Requirements:

2. Please ensure that the Title in your manuscript file and the Title provided in your online submission form are the same.

Additional Editor Comments (if provided):

Reviewers' comments:

Reviewer's Responses to Questions

**Comments to the Author**

1. If the authors have adequately addressed your comments raised in a previous round of review and you feel that this manuscript is now acceptable for publication, you may indicate that here to bypass the “Comments to the Author” section, enter your conflict of interest statement in the “Confidential to Editor” section, and submit your "Accept" recommendation.

Reviewer #1: All comments have been addressed

Reviewer #2: (No Response)

2. Does this manuscript meet PLOS Global Public Health’s publication criteria? Is the manuscript technically sound, and do the data support the conclusions? The manuscript must describe methodologically and ethically rigorous research with conclusions that are appropriately drawn based on the data presented.

Reviewer #1: Yes

Reviewer #2: Yes

3. Has the statistical analysis been performed appropriately and rigorously?

Reviewer #1: N/A

Reviewer #2: N/A

4. Have the authors made all data underlying the findings in their manuscript fully available (please refer to the Data Availability Statement at the start of the manuscript PDF file)?

Reviewer #1: No

Reviewer #2: Yes

5. Is the manuscript presented in an intelligible fashion and written in standard English?

Reviewer #1: Yes

Reviewer #2: Yes

6. Review Comments to the Author

Reviewer #1: A minor clarification though:

Page6: Research design is a qualitative phenomenological, but does not mention the techniques for data collection; however, on page 10 sub-title Fear of physical assault for family…- Line 5 it is mentioned- in depth interviews with SMM, kindly clarify in the research design whether in depth interview only data collection technique.

Reviewer #2: The authors have addressed most of the key comments and the revisions have strengthened the manuscript.

Family: In the Discussion section, the authors can discuss the possibility that SMM may be concerned that their family members might be harmed by others/society if they disclose their sexual orientation (an alternative/rival explanation or evidence for which may be available in their data). The findings only argue that SMM might be harmed by their family, so they do not disclose their sexual orientation to the family (or others). The authors can briefly speculate about the potential influence of religion on the family's attitudes, whether or in what ways the religion (or other factors) contributed to the proposed legislation, and the potential influence of the proposed legislation on the family’s attitudes toward SMM.

Regarding the type of sampling: It appears to be convenience sampling as the authors recruited SMM from two slums and they did not use any time slots for recruitment. The authors need to explain why they consider it as a time-location sampling. If not, they need to revise the sentence on the type of sampling – retaining the term ‘convenience sampling’ but deleting the term “time-location sampling" or "TLS”.

7. PLOS authors have the option to publish the peer review history of their article (what does this mean?). If published, this will include your full peer review and any attached files.

**Do you want your identity to be public for this peer review?** For information about this choice, including consent withdrawal, please see our Privacy Policy.

Reviewer #1: **Yes: **Emmy Kageha Igonya

Reviewer #2: No

---

## [Editor Report · Decision Letter 2]

25 Oct 2023

PGPH-D-23-00095R2

Family rejection of non-heterosexuality – sexual orientation and behavior anonymity among sexual minority men in slum communities-BSGH 001.

Dear Dr. Shamrock,

Thank you for submitting your manuscript to PLOS Global Public Health. After careful consideration, we feel that it has merit but does not fully meet PLOS Global Public Health’s publication criteria as it currently stands. Therefore, we invite you to submit a revised version of the manuscript that addresses the points raised during the review process.

You have addressed all the additional points raised, however there are corrections needed in two of the revised sentences in the text due to grammatical errors and redundancy.

1. It is preferred in social research not to speak of “proof” but rather something like evidence or support.

p. 22. Change this:

“While we acknowledge with proof that SMM avoided disclosing their sexual orientation/behavior to family due to the perceived risk of facing harm from their relatives. We also recognize that SMM may avoid disclosure in their communities to prevent attacks on their family members (Boulder, 2018; Lavietes, 2022).”

To something like this:

While we identified evidence that SMM avoided disclosing their sexual orientation/behavior to family due to the perceived risk of facing harm from their relatives, we also recognize that SMM may avoid disclosure in their communities to prevent attacks on their family members (Boulder, 2018; Lavietes, 2022).

2.  Since you did not identify this in your own study, it is best to use more tentative wording (i.e., not “urge”); and the first phrase is unnecessary: Delete: “While these considerations are being considered….”  Also, by protections do you mean resources to assist families? 

p. 27: Change this:

“While these considerations are being considered, we also urge for strategies to consider and place protections for families who are in support of their relatives’ sexual orientation/behavior.”

To something like this:

We also suggest strategies to consider and implement resources for families who are in support of their relatives’ sexual orientation/behavior.

We look forward to receiving your revised manuscript.

Kind regards,

Peter A. Newman, Ph.D

Academic Editor
---

## [Editor Report · Decision Letter 3]

8 Nov 2023

Family rejection of non-hetero sexuality – sexual orientation and behavior anonymity among sexual minority men in slum communities-BSGH 001.

PGPH-D-23-00095R3

Dear Dr Shamrock,

We are pleased to inform you that your manuscript 'Family rejection of non-hetero sexuality – sexual orientation and behavior anonymity among sexual minority men in slum communities-BSGH 001.' has been provisionally accepted for publication in PLOS Global Public Health.

Best regards,

Peter A. Newman, Ph.D

Academic Editor